# Low Free Triiodothyronine as a More Sensitive Predictor of Survival Than Total Testosterone among Dialysis Men

**DOI:** 10.3390/nu15030595

**Published:** 2023-01-23

**Authors:** Ksymena Leśniak, Aleksandra Rymarz, Maria Sobol, Stanisław Niemczyk

**Affiliations:** 1Department of Internal Diseases, Nephrology and Dialysis, Military Institute of Medicine—National Research Institute, 04-141 Warsaw, Poland; 2Department of Biophysics and Human Physiology, Medical University of Warsaw, 02-004 Warsaw, Poland

**Keywords:** fT3, survival, total testosterone, dialysis

## Abstract

Background: Some endocrine disorders, previously considered benign, may be related to a poorer prognosis for patients with renal failure. Both low serum free triiodothyronine (fT3) and low total testosterone (TT) concentrations have been considered as predictors of death in dialysis patients, but the results of studies are inconsistent. In our study, we evaluated the relationships of the serum thyroid hormone levels and the total testosterone levels with survival in male dialysis patients. Methods: Forty-eight male dialysis patients, 31 on hemodialysis (HD) and 17 on peritoneal dialysis (PD), aged 61.4 ± 10.0, 59.2 ± 12.2 years, respectively, were included in the study. Serum thyroid hormones and total testosterone were measured. Results: During the 12-month follow-up, nine all-cause deaths were recorded. The concentrations of fT3 were significantly lower in those who died than in the survivors (*p* = 0.001). We did not observe any statistically considerable differences between the group of men who died and the rest of the participants in terms of the total serum testosterone concentration (*p* = 0.350). Total testosterone positively correlated with fT3 (r = 0.463, *p* = 0.009) in the HD group. Conclusions: In the group of male dialysis patients, the serum concentration of fT3 had a better prognostic value in terms of survival than the total testosterone. A linear relationship between the fT3 levels and testosterone levels in men undergoing hemodialysis may confirm the hypothesis that some of the hormonal changes observed in chronic kidney disease (CKD) may have a common cause.

## 1. Introduction

The kidneys are a valid endocrine organ, a pivotal regulator of endocrine functions, not only by producing hormones, but also by influencing their metabolism and elimination [1]. Patients suffering from kidney failure often have thyroid dysfunction (low T3 syndrome) and testosterone deficiency (hypogonadism), which are known to be predictors of poor clinical outcomes [2,3,4,5,6,7,8,9].

Thyroid hormones change as kidney function deteriorates. Low T3 syndrome is characterized by reduced total triiodothyronine (T3) and free triiodothyronine (fT3) in the presence of normal thyroid stimulating hormone (TSH) and thyroxine (T4) [9]. According to reports, as many as 80% of patients with end-stage renal disease (ESRD) have a low concentration of T3 [10].

The underlying mechanism of derangement in the thyroid hormone levels and/or metabolism in CKD is caused by both general and kidney-specific factors such as reduced bioactivities of thyrotropin-releasing hormone (TRH) and TSH, loss of thyroid hormone binding proteins during dialysis, modified entry of the thyroid hormone into tissue, altered expression of iodothyronine deiodinases, changes in thyroid hormone receptor expression or function, iodide retention, anemia, and comorbidities associated with CKD [11,12]. It should be mentioned that medications that are often prescribed for patients with CKD (corticosteroids, amiodarone, propranolol, and lithium) can also inhibit the metabolism of thyroid hormones [13,14]. Many CKD patients have low fT3 levels due to decreased peripheral T4 to T3 conversion. Reduced peripheral conversion results from starvation, protein deficiency, and metabolic acidosis, which impair iodothyronine deiodination and T3 binding with proteins [15]. Inflammatory mediators including tumor necrosis factor-TNF alpha and IL1, the concentration of which is elevated in CKD, also reduce the activity of 1′ 5′ deiodinase [15]. It is possible that testosterone and selenium deficiency may also be involved in impaired deiodinase activity [16,17].

Low T3 syndrome is not typical only for CKD, but also for infectious, neurological, cardiovascular, or hematologic diseases. Traditionally, low T3 syndrome has been considered as a way to preserve energy in response to wasting in severe illness such as CKD [18]. However, over the past 20 years, there has been evidence of the role of T3 in the pathophysiology of endothelial dysfunction, atherosclerosis, and cardiac abnormalities in the general population [19,20].

Accordingly, in recent years, intriguing and consistent associations have been reported regarding T3 levels and inflammation, endothelial disruption, arterial stiffness, and cardiomyopathy in end-stage renal disease patients and hemodialysis patients [3,9,21,22,23,24,25]. Moreover, several observational studies have demonstrated that single measurements of T3 levels are independent predictors of all-cause and cardiovascular mortality in patients suffering from advanced renal failure and those undergoing dialysis [3,4,5,9,26]. However, not all research has shown that T3 levels are an independent predictor of death [27,28,29]. A recently published study performed on a cohort of 438 hemodialysis patients in Japan (DREAM) revealed an association between low fT3 concentration and the increase in both mortality and the prevalence of new cardiovascular events; however, this relationship disappeared after the adjustment for the history of cardiovascular diseases and the concentration of NT-pro-BNP [29]. In turn, in the study conducted by Ozen et al., the authors showed that the relationship between fT3 and mortality was disrupted by the nutritional status of the hemodialysis patients [27]. Fernández-Reyes et al., in a smaller study of 89 hemodialysis patients, found no differences in the fT3 concentrations between the group of deceased patients and those who survived. Furthermore, this Spanish group of scientists indicated that the fT3 concentration does not predict long-term survival [28].

It is worth mentioning that Xu et al. in the meta-analysis including twelve larger studies with a number of 14,766 dialysis patients revealed that low levels of fT3 and free thyroxine (fT4) were associated with higher risk of death in this cohort [30].

Most publications on the associations between thyroid hormones and mortality in the CKD concerned groups of patients of both sexes [3,4,5,9,26,27,28,29]. However, as previously reported, the serum total testosterone levels in men undergoing dialysis may also have prognostic significance. The incidence of hypogonadism (testosterone deficiency) in men receiving dialysis was significantly higher than in the general population, ranging from approximately 40% in the Japanese population, 50% in the American population, and 45–65% in the European population [2,31,32,33]. Deficiency of total testosterone, which is a potent anabolic hormone, is associated with the onset of muscle wasting (or sarcopenia) [34], anemia [35], skeletal demineralization [36], malnutrition [37,38], and frailty syndrome [39] in patients with CKD. What is extremely important is that there is increasing evidence that testosterone deficiency is associated with greater cardiovascular risk and mortality in this population [2,6,7,8,32,40,41,42], but not all studies have revealed that testosterone is an independent predictor of mortality [8,32].

Our objective was to evaluate the associations of the serum thyroid hormone levels and total testosterone levels with survival in male dialysis patients.

## 2. Materials and Methods

This is an observational, prospective study performed at the Department of Internal Diseases, Nephrology and Dialysis Therapy of the Military Institute of Medicine—National Research Institute in Warsaw.

The study included forty-eight male patients aged between 41 and 84 years: 31 on hemodialysis (HD) and 17 on peritoneal dialysis (PD). We received written informed consent from all patients participating in the study. 

We included adult male patients with end-stage renal disease treated with hemodialysis or peritoneal dialysis.

We excluded patients with a dialysis vintage less than 3 months, active malignancy, hormone supplementation for endocrine diseases (including androgen supplementation, levothyroxine replacement, treatment with corticosteroids, thiamazole, or propylthiouracil) and severe clinical condition including the active inflammatory state and severe liver disease.

The study protocol was reviewed and approved by the Military Institute of Medicine Bioethics Committee (approval number 50/WIM/2016). The study was conducted in accordance with the guidelines of the Declaration of Helsinki. 

At the beginning of the study, the demographic characteristics and medical history were collected from each participant. The concentrations of TSH, fT3, fT4 and the total testosterone were measured in the serum. The laboratory parameters were measured at the study entry and after the 12-month follow up.

Blood samples were collected between 7.00– and 11.00 and analyzed immediately after collection.

All laboratory measurements were performed by the Department of Laboratory Diagnostics of the Military Institute of Medicine–National Research Institute, Warsaw.

Serum TSH (normal range: 0.27–4.2 uIU/l), fT3 (normal range: 3.2–6.9 pmol/l), fT4 (normal range: 12–22 pmol/l), and total testosterone (normal range: 2.8–8.2 ng/mL) were determined by the electrochemiluminescence method (Roche Elecsys analyzer, Mannheim, Germany). 

Low-T3 syndrome was considered if the serum fT3 level was below the reference range, the TSH level was within the normal range, and the fT4 level was normal or low [43].

Cardiovascular diseases (CVD) included heart failure, coronary artery disease, heart infarct, ischemic stroke, intracerebral hemorrhage, and significant peripheral arterial disease of the lower limbs.

After reviewing the available medical information, all-cause mortality as well as the date and cause of death were summarized. Cardiovascular (CV) death was determined as death caused by sudden cardiac death, ischemic heart disease, stroke, and peripheral arterial disease of the lower limbs [8].

### Statistical Analysis

A statistical analysis was performed using the Statistical13 package. Biographical information and quantitative variables were shown with the use of descriptive statistics (mean, standard deviation, median, and range). The distribution of each quantitative variable was checked with the normal distribution (Shapiro–Wilk test). The required sample size was calculated using two-sample t-test power analysis. 

The qualitative variables were summarized as the percentage distribution. Nominal variables were analyzed by the Chi-square test. The nonparametric Mann–Whitney test was performed to evaluate the differences between the HD and PD groups. Correlation analysis was conducted using the Spearman or the Pearson correlation coefficient. The multivariate logistic regression was used to evaluate the risk factors of mortality. The results were shown as the odds ratio (OR) with a 95% confidence interval (CI). The receiver-operating characteristic (ROC) analysis was conducted to discern the best cut-off point value of the serum total testosterone concentration, TSH concentration, or serum fT3 and fT4 concentration in the male dialysis group. The area under the curve (AUC) was calculated for each considered parameter. The results were considered as statistically significant if the *p* value was less than 0.05.

## 3. Results

### 3.1. General Characteristics

A total of 48 male dialysis patients (31 on HD, 17 on PD) were included in the study. The baseline characteristics of the study groups are detailed in Table 1.

Low T3 syndrome was found in 16.1% of hemodialysis patients and 5.8% of peritoneal dialysis patients. 

We found statistically significant differences between the groups for the following parameters: duration of dialysis and the concentration of fT4 (Table 1).

On the other hand, there was a statistically significant decrease in the fT4 and fT3 concentrations after 12 months in the HD group with *p* values of 0.009 and <0.001, respectively.

### 3.2. Correlation Analysis

Table 2 shows the correlations between the thyroid hormone and other variables in the HD group. 

Free triiodothyronine correlated positively with the total testosterone concentration in the HD group (*p* = 0.009). However, we did not observe any statistically significant correlations between these parameters in the PD group.

### 3.3. Implications on Outcome

During the 12-month follow-up period, nine (18.7%) patients died (five individuals died in the HD group and four individuals died in the PD group). CV deaths were recorded in 8/9 deceased men, one person died from systemic infection. All deceased patients had prior CVD, and 8/9 of the dead patients were diabetics. Other co-morbidities in the group of deceased patients were as follows: anemia (7/9), secondary hyperparathyroidism (4/9), atrial fibrillation (1/9), venous/pulmonary thromboembolism (2/9), gastritis (2/9), gout (1/9), benign prostatic hyperplasia (2/9), osteoarthritis (1/9), cataract (3/9), glaucoma (1/9), gallbladder stone (2/9), and epilepsy (1/9).

The comparison between the group of men who died and the rest of the patients is included in Table 3. 

The nine deceased men had a statistically lower concentration of fT3 in the serum (*p* = 0.001) compared with the remaining 39 men (Figure 1).

The analysis of the ROC curves showed the significance of the fT3 level in the serum as a prognostic marker in the male dialysis patients (*p* < 0.001); (OH:AUC = 0.879; 95% CI: 0.753–1.0) (Table 4). The suggested cut-off value of the fT3 concentration to determine the prognosis is 3.88 pmol/l. It is worth noting that the concentration of fT3 was a better prognostic marker than the concentration of total testosterone in the serum (Figure 2).

In order to predict mortality, a multivariate logistic regression model was applied including age and duration of the dialysis treatment. None of the considered independent variables were statistically significant, the received odds ratio (OR) for age was 1.011 with CI of 0.943 to 1.084 (*p* = 0.751), and the duration of dialysis OR = 1.019 with a CI of 0.758 to 1.371 (*p* = 0.900).

In the HD group, the Kaplan–Meier analysis for all-cause mortality according to the presence of cardiovascular diseases and diabetes mellitus revealed no statistically significant difference in survival for any distribution of the subjects (*p* = 0.923 according to the cardiovascular disease and *p* = 0.087 according to the presence of diabetes, respectively). 

## 4. Discussion

In our study, we evaluated both the hormonal predictors of mortality in male dialysis patients: fT3 and the total testosterone serum concentrations. 

Based on our findings, low fT3 level was related to an increased risk of all-cause mortality in a cohort of male dialysis patients, which is consistent with previous observations [3,4,5,9]. 

Zoccali et al. revealed a direct relationship between low fT3 and all-cause mortality in a group of 200 individuals on maintenance hemodialysis [3]. Moreover, Meuwese et al. assessed the relationship between T3 (but not fT3) variability and mortality in a cohort of predominantly hemodialysis patients. They observed the highest risk of death in patients with persistently low T3 concentrations (HR: 2.7; 95% IC 1.5-5.0) compared to patients with persistently high concentrations [4]. On the other hand, Chang et al., in the prospective observational study including 447 euthyroid patients treated with peritoneal dialysis, demonstrated that T3 concentration at the onset of PD was a strong independent predictor of long-term cardiovascular mortality [5].

There is a diverse spectrum of abnormalities in the thyroid hormone levels in chronic kidney disease. As a consequence, various studies might differ in the thyroid hormone values used to determine low T3 syndrome. Moreover, there is no strictly defined cut-off value of fT3 to assess the prognosis in patients with CKD, therefore, we compared the results of our study with other researchers.

In our study, the group of nine deceased men (3.1 ± 0.8 pmol/l = 2.0 ± 0.5 pg/mL) had a statistically lower concentration of fT3 in the serum (*p* = 0.001) compared with the remaining 39 men (4.2 ± 0.7 pmol/l = 2.7 ± 0.4 pg/mL). We also suggest that the cut-off value of fT3 for the determining of prognosis in dialysis patients is 3.88 pmol/l = 2.52 pg/mL.

For comparison, Zoccali et al., in an Italian study, showed that participants who were on persistent hemodialysis (3.3 ± 0.8 pg/mL) had lower fT3 (*p* < 0.001) than the healthy subjects (3.7 ± 1.0 pg/mL) and clinically euthyroid patients with normal renal function (3.6 ± 0.8 pg/mL) [23]. This author, in another study, demonstrated that the hemodialysis participants who died (3.1 ± 0.7 pg/mL) had a significantly lower serum fT3 level (*p* < 0.001) than those who survived (3.7 ± 0.8 pg/mL). Moreover, this analysis showed that an increase in the serum fT3 concentration by 1 pg/mL was associated with a 50% decrease in the risk of all-cause death (HR: 0.50, 95% CI: 0.35–0.72; *p* < 0.001) [3].

In turn, Enia et al., in a study conducted on 41 peritoneal dialysis patients in Italy reported that PD patients had lower serum fT3 concentrations (2.7 ± 0.8 pg/mL) than similarly aged healthy participants (3.7 ± 1.0 pg/mL, *p* < 0.001). In addition, a lower concentration of fT3 was found in the group of deceased PD patients (2.5 ± 0.8 pg/mL) in comparison to the group of PD patients who survived (3.3 ± 0.5 pg/mL, *p* = 0.001) [26].

In a Turkish study of 137 hemodialysis patients, Tatar et al. showed that patients in the low fT3 tertile (fT3 <3.27 pg/mL) were older, had lower serum hemoglobin and albumin concentrations, and had higher hs-CRP levels compared with high fT3 tertiles (fT3 >3.85 pg/mL) [21]. 

In a post hoc analysis of a cohort of 210 nondialysed CKD stage 5 participants, authors from Sweden noted that both low plasma T3 and low plasma fT3 had an impact on mortality, but only after including all patients (euthyroid and noneuthyroid). Interestingly, the predictive values of T3 and fT3 in this study were similar and amounted to 1.9 pg/mL [9].

Undoubtedly, changes in the levels and function of thyroid hormones have adverse effects on the cardiovascular system [15].

Recently, Mastorici et al. reported detailed experimental findings showing the cardioprotective effects of thyroid hormones. The authors of this study presented the influence of thyroid hormones on pathways (epigenetic modification, cell growth and differentiation, mitochondrial functioning, apoptosis, myocardial structure, neoangiogenesis and fibrosis, oxidative stress, and inflammation), which may be the targets of thyroid hormone supplementation therapy [15].

The influence of the thyroid dysfunction on the mortality of CKD patients is still unknown, but may be partly explained by its important association with the nutritional state and inflammation [3,23,27]. 

Ozen et al., in a large study of 669 hemodialysis patients, indicated that the fT3 level is a strong predictor of overall mortality. However, after including serum albumin and hs-CRP concentrations (bioindicators of malnutrition and inflammation) in the analysis, the association of fT3 with mortality disappeared [27]. Carrero et al., in a study of patients with nondialysed ESRD, found a relationship between thyroid hormones and parameters of wasting and inflammation. They showed that T3 and fT3 negatively correlated with IL-6, hs-CRP, and vascular adhesion molecule (VCAM)-1 and positively correlated with s-albumin and IGF-1 [9]. Similar correlations were presented by Enia et al. in the study concerning the peritoneal dialysis patients. The authors of this research suggest a possible involvement of the malnutrition-inflammation syndrome in the development of the low T3 syndrome in these patients [26].

Perhaps inflammation mediates between low fT3 and left ventricular dysfunction/hypertrophy in patients with ESRD [24].

A strong negative correlation has been shown between low fT3 with markers of inflammation (Interleukin-6, C-reactive protein) and endothelial activation (intercellular adhesion molecule-1 [ICAM-1], VCAM-1) in stable patients with ESRD [23].

In the available literature, there are reports of the relationship between the concentration of free triiodothyronine in patients with ESRD and artery calcification [21,44], intima-media thickness [21], and measures of systemic arterial stiffness [21,44,45]. Several potential mechanisms might enhance this association. The ex-vivo studies have revealed that low levels of free triiodothyronine reduce the synthesis of klotho and matrix Gla protein, thereby increasing vascular calcification [46,47].

Low fT3 in CKD patients may be a marker of endothelial dysfunction, which is the underlying disorder leading to vascular destruction in CKD [22,48]. 

Endothelial dysfunction is diagnosed by measuring endothelial markers or blood flow in the forearm (FBF) response to ischemia. A positive correlation was observed between fT3 concentration and FMD (flow- mediated vasodilatation) in patients with CKD [22]. Yilmaz et al. showed that low fT3 concentrations were associated with an impaired response of blood flow in the forearm to ischemia in the cohort of 217 patients with CKD stages 3–4 without diabetes and in the clinical euthyroid state. This association was independent of the classical risk factors and risk factors specific to CKD such as glomerular filtration rate (GFR), proteinuria, phosphate, or hemoglobin serum concentrations [22].

Including the above considerations, it is natural to raise the question about the importance of treating thyroid hormone disorders in patients with CKD and on dialysis. As above-mentioned, interactions between the kidneys and the thyroid gland occur at many levels and develop through many mechanisms. These impacts are not only functional but structural. It should be noted that in CKD, the results of the hormonal tests must be carefully and judiciously interpreted [11,12].

Low T3 syndrome without an increase in rT3, elevated TSH and low fT4, and impaired TSH response to TRH are disorders found in CKD where supplementary thyroid hormone therapy could potentially benefit, despite the supposed “euthyroidism” [49]. However, despite many years of research in this field, thyroid hormone supplementation in CKD remains debatable. It is even believed that such substitution therapy may have an adverse effect on the patient’s condition. Attempts to supplement T3 led to a negative nitrogen balance resulting from the excessive catabolism of muscle proteins in patients with CKD. The degree of thyroid hormone abnormalities, which is the threshold for thyroxine substitution therapy in CKD, is also not specified [11,49]. In general, it is assumed that a mild rise in TSH to less than 20 IU/mL with or without low T3/T4 does not require absolute thyroid hormone supplementation in CKD [49]. Both the risks of hyperthyroidism and the risks and benefits of hypothyroidism must be considered. Treatment decisions should be made on an individual patient basis after weighing the likely benefits and potential risks of thyroid hormone therapy or no pharmacological intervention [49]. 

There were also some differences in the dialysis patients. Heparin administered for hemodialysis procedures inhibits the binding of T4 to proteins, thus increasing the fraction of free T4. In addition, hemodialysis patients have a compensatory effect on the cellular transport of thyroid hormones, which helps maintain euthyroidism despite the low serum thyroid hormone levels. Taking this into consideration, it is believed that despite the low serum thyroid hormone profile, thyroid hormone supplementation should not be initiated without a significant increase in the TSH levels and careful consideration in hemodialysis patients. Thyroxine-binding globulin (TBG) and T3 and T4 are lost during peritoneal dialysis. Despite the significant loss of TBG, the T3 and T4 losses were easily compensated and their concentrations were kept within the normal limits. Patients treated with peritoneal dialysis do not therefore require substitution with thyroid hormones [49]. 

A team of nephrologists and endocrinologists should jointly determine strategies for the treatment of patients with CKD and thyroid hormone abnormalities, taking into account the individualization of therapy in relation to the needs and clinical condition of the individual patients.

It is worth noting that in most studies on the relationship between thyroid hormone test disorders and mortality in patients with ESRD undergoing dialysis, the study group consisted of both men and women [3,4,5,9,26,27,28,29]. Therefore, we included only male dialysis patients in our study, in whom we assessed, in addition to thyroid hormones, the concentration of total testosterone, which is also a potential predictor of mortality in this group of patients [6,7,8,40,41,42].

In contrast to our predictions, no relationship between the concentration of total testosterone in the serum and mortality in male dialysis patients was observed. 

Referring to the literature data, it should be noted that the results of the studies on the relationship between the serum total testosterone and mortality in patients with CKD are inconsistent. 

Carrero et al. performed a study of 126 Swedish hemodialysis patients, which demonstrated that those with the lowest total testosterone concentration compared to the group with the highest concentration had a higher risk of death regardless of the socio-demographic data, comorbidities, medications, albumin levels, and inflammatory parameters; however, after adjusting for serum creatinine as an indicator of muscle mass, these associations became insignificant [8]. Similarly, in the Canadian study, Bello et al. reported a statistically significant trend of a higher risk of cardiovascular events among participants with low serum testosterone levels (*p* < 0.001), but this risk was no longer significant after adjusting for age [42].

However, in our study, the total testosterone positively correlated with fT3 in the group of male hemodialysis patients (r = 0.463, *p* = 0.009). 

A similar correlation was previously noticed in the paper by Mauwese et al. The authors presented an interesting hypothesis concerning the relationship between hormonal disorders of the hypothalamic–pituitary–organs axis and inflammation in CKD. The authors suggest that some of the hormonal changes observed in CKD may have a common causation in the shape of the downregulation of the hypothalamus and pituitary gland. In an endorsement, they presented a linear relationship between the T3 and testosterone levels in men undergoing hemodialysis [50]. 

The authors hypothesized that the weakening of this axis could mean the activation of a “safe mode” in which the metabolism was lowered and the unnecessary functions were damped. Systemic inflammation associated with CKD may play a substantial role in this process [51]. The release of pro-inflammatory cytokines, which can partially suppress the hypothalamic–pituitary–gonadal axis [52,53,54] or reduce the conversion of T4 to T3 [55], has a temporarily beneficial effect on survival; however, the chronic process can be harmful. 

It is worth mentioning that inflammation is a component of malnutrition-inflammation-atherosclerosis syndrome which is considered a relevant cause of all-cause and cardiovascular mortality in dialysis patients [56].

Perhaps the concomitant hormonal disorders of the thyroid (low fT3 level) and the gonads (testosterone deficiency), together with microinflammation and malnutrition, are in the common pathway leading to death in the group of male patients with advanced renal failure. Further research exploring these interesting issues are therefore needed.

The limitation of our study was the small sample size and relatively short period of observation. Nevertheless, the relationship between low fT3 and survival in chronic kidney disease patients follows the trends also observed by other researchers. 

## 5. Conclusions

Thyroid hormone derangement, in particular low fT3 level, appears to contribute to increased mortality in dialysis patients. Unfortunately, it is not yet clear whether correcting abnormal thyroid function will favorably influence mortality in this cohort. 

## Figures and Tables

**Figure 1 nutrients-15-00595-f001:**
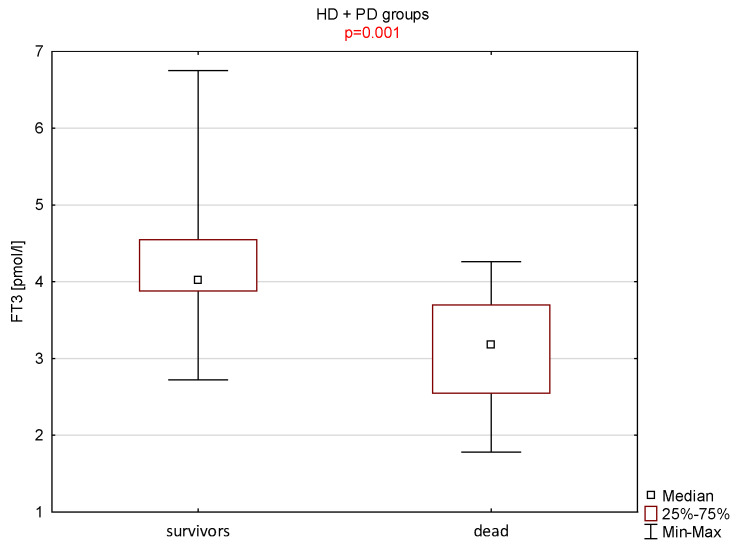
The differences between the group of men who died and the rest of the patients in terms of the serum fT3 concentrations.

**Figure 2 nutrients-15-00595-f002:**
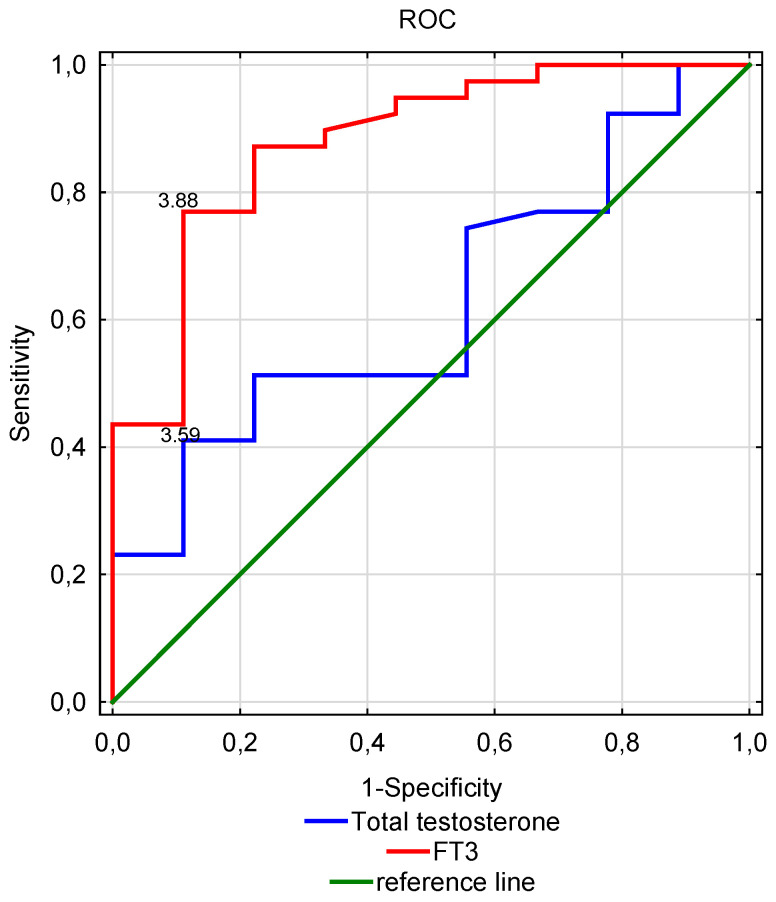
ROC analysis of fT3 and the total testosterone serum concentrations in the determination of prognosis in male dialysis patients.

**Table 1 nutrients-15-00595-t001:** Evaluation of the clinical and hormonal profile in the studied groups of men.

	HD	PD	*p* *
	N = 31	N = 17	
Age (years, range)	61.4 ± 10.0; 42–84	59.2 ± 12.2; 41–77	0.606
Duration of dialysis (years, range)	2.8 ± 2.7; 0.3–12.8	1.7 ± 1.6; 0.3–6.8	0.010
Prior CVD, %	74.2%	64.7%	0.534
DM, %	61.3%	47.0%	0.391
Hypertension, %	100.0%	100.0%	1
Total testosterone (ng/mL, range)	3.1 ± 1.3; 1.1–8.0	3.7 ± 1.3; 1.7–6.4	0.070
TSH (uIU/mL, range)	1.7 ± 1.1; 0.4–5.6	2.2 ± 1.2; 0.8–5.7	0.132
fT3 (pmol/L, range)	4.0 ± 0.9; 1.8–6.8	4.0 ± 0.7; 2.1–4.9	0.608
fT4 (pmol/L, range)	14.8 ± 2.4; 11.3–20.8	16.4 ± 2.4; 13.3–20.7	0.038

* Overall comparison in the Mann–Whitney test. CVD = cardiovascular diseases, DM = diabetes mellitus.

**Table 2 nutrients-15-00595-t002:** Correlations between the thyroid hormone concentrations and age, total testosterone concentrations, and dialysis duration in the HD group.

		Age	TT	Dialysis Duration
TSH	r	−0.175	−0.068	−0.147
	*p*	0.347	0.717	0.429
fT3	r	0.053	0.463	−0.029
	*p*	0.779	0.009	0.877
fT4	r	−0.038	−0.073	−0.356
	*p*	0.841	0.702	0.054

TT = total testosterone.

**Table 3 nutrients-15-00595-t003:** Comparison between the group of men who died and the rest of the male dialysis patients in terms of age, dialysis duration, and the concentration of hormones in the blood serum.

Parameter	Non Survivors			Survivors			*p*-Value *
	n	Mean ± SD	Median	n	Mean ± SD	Median	
Age (years)	9	62 ± 11	59	39	60 ± 11	63	0.728
Dialysis duration (years)	9	2.5 ± 2.2	1.6	39	2.3 ± 2.5	1.6	0.755
Total testosterone (ng/mL)	9	2.8 ± 0.9	3.1	39	3.4 ± 1.4	3.3	0.350
TSH (uIU/mL)	9	2.3 ± 1.5	2.1	39	1.8 ± 1.1	1.7	0.348
fT3 (pmol/L)	9	3.1 ± 0.8	3.2	39	4.2 ± 0.7	4.0	0.001
fT4 (pmol/L)	9	16.1 ± 2.6	16.2	39	15.2 ± 2.4	15.1	0.395

* Overall comparison in the Mann–Whitney test.

**Table 4 nutrients-15-00595-t004:** ROC analysis of the selected hormone serum concentrations in the determination of prognosis in male dialysis patients.

Parameter	Cut-Off Value	Sensitivity	Specificity	AUC	Significance-*p*
Total testosterone (ng/mL)	3.59	0.41	0.12	0.625	0.188
TSH (uIU/mL)	2.23	0.77	0.56	0.603	0.326
fT3 (pmol/L)	3.88	0.77	0.11	0.879	<0.001
fT4 (pmol/L)	15.54	0.56	0.25	0.599	0.375

## Data Availability

The data presented in this study are available on request from the corresponding author. The data are not publicly available due to privacy.

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
