# Peer review of "Low Free Triiodothyronine as a More Sensitive Predictor of Survival Than Total Testosterone among Dialysis Men"

_nutrients, 2023, doi:10.3390/nu15030595_

Round 1
Reviewer 1 Report
A study was done to demonstrate a correlation between testosterone and Triiodothyronine with dialysis in men. Authors found that lower levels of Triiodothyronine are associated with poor prognosis. Overall, the data supports their conclusion, and the methods are described well. Previous studies have already shown that kidney defects result in the deterioration of thyroid hormones. This particular study confirms this in patients that were admitted to one hospital in Poland. Hence, the overall impact and novelty are limited. Some suggestions to improve the manuscript.
1) It is suggested to change the title into a statement rather than a question.
2) Add Triiodothyronine levels of control/healthy parties that belong to the same ethnicity
3) Mention any co-morbidities that the patients have especially fr the one that died
Reviewer 2 Report
By using appropriate statistical analysis, this manuscript illustrates that low free triiodothyronine acts as a more sensitive predictor of survival than total testosterone in men on dialysis. In general, the trial design and statistical methods of this cohort study are scientific and reasonable. Apart from this, I have a few minor issues that need to be addressed:
MINOR CONCERNS
1) The manuscript provided a detailed description of the association between T3 and CKD in the background section, but did not provide an adequate description of the association between total testosterone levels and dialysis patients.
2) The exclusion criteria are described in detail in the text. The inclusion criteria should be appropriately described on this basis.
3) I suggest that the discussion section should include an exploration of the potential mechanisms by which thyroid hormones affect death in dialysis patients.
4) The manuscript did not mention the calculation of sample size, I recommend that the author present the result based on statistics.
5) I recommend streamlining the number of keywords to less than 5.
